# Asthma: New Integrative Treatment Strategies for the Next Decades

**DOI:** 10.3390/medicina56090438

**Published:** 2020-08-28

**Authors:** Diego A. Arteaga-Badillo, Jacqueline Portillo-Reyes, Nancy Vargas-Mendoza, José A. Morales-González, Jeannett A. Izquierdo-Vega, Manuel Sánchez-Gutiérrez, Isela Álvarez-González, Ángel Morales-González, Eduardo Madrigal-Bujaidar, Eduardo Madrigal-Santillán

**Affiliations:** 1Instituto de Ciencias de la Salud, Universidad Autónoma del Estado de Hidalgo, Ex-Hacienda de la Concepción, Tilcuautla, Pachuca de Soto 42080, Mexico; diego060195@hotmail.com (D.A.A.-B.); jacke_star230990@hotmail.com (J.P.-R.); jizquierdovega@gmail.com (J.A.I.-V.); spmtz68@yahoo.com.mx (M.S.-G.); 2Escuela Superior de Medicina, Instituto Politécnico Nacional, “Unidad Casco de Santo Tomas”, Ciudad de México 11340, Mexico; nvargas_mendoza@hotmail.com (N.V.-M.); jmorales101@yahoo.com.mx (J.A.M.-G.); 3Escuela Nacional de Ciencias Biológicas, Instituto Politécnico Nacional, “Unidad Profesional A. López Mateos”, Ciudad de México 07738, Mexico; isela.alvarez@gmail.com (I.Á.-G.); eduardo.madrigal@lycos.com (E.M.-B.); 4Escuela Superior de Cómputo, Instituto Politécnico Nacional, “Unidad Profesional A. López Mateos”, Ciudad de México 07738, Mexico; anmorales@ipn.mx

**Keywords:** asthma, diagnosis, treatment strategies, oxidative stress, antioxidants

## Abstract

Asthma is a chronic disease whose main anatomical–functional alterations are grouped into obstruction, nonspecific bronchial hyperreactivity, inflammation and airway remodeling. Currently, the Global Initiative of Asthma 2020 (GINA 2020) suggests classifying it into intermittent cases, slightly persistent, moderately persistent and severely persistent, thus determining the correct guidelines for its therapy. In general, the drugs used for its management are divided into two groups, those with a potential bronchodilator and the controlling agents of inflammation. However, asthmatic treatments continue to evolve, and notable advances have been made possible in biological therapy with monoclonal antibodies and in the relationship between this disease and oxidative stress. This opens a new path to dietary and herbal strategies and the use of antioxidants as a possible therapy that supports conventional pharmacological treatments and reduces their doses and/or adverse effects. This review compiles information from different published research on risk factors, pathophysiology, classification, diagnosis and the main treatments; likewise, it synthesizes the current evidence of herbal medicine for its control. Studies on integrative medicine (IM) therapies for asthmatic control are critically reviewed. An integrative approach to the prevention and management of asthma warrants consideration in clinical practice. The intention is to encourage health professionals and scientists to expand the horizons of basic and clinical research (preclinical, clinical and integrative medicine) on asthma control.

## 1. Introduction

Asthma is a syndrome that can present different superimposed phenotypes with defined clinical and physiological characteristics, and with sub-adjacent inflammatory processes with identifiable biomarkers whose risk factors can be genetic, environmental, and/or represent the interaction of both of these [1]. Due to its alterations in respiratory function, its different clinical expressions (which can vary with age at presentation), and its presenting a multifactorial etiology, it is complicated to find an exact definition that fully describes this entity. The Global Initiative for Asthma 2006 (GINA 2006) considers the functional as well as the cellular aspect, and proposed its definition as a chronic inflammatory disease of the airways in which diverse cells and cellular products play an important role [2].

In general, asthma is characterized by a chronic inflammation of the airways, which is distinguished by classic respiratory symptomology that encompasses wheezing, respiratory difficulty (principally at night or during the early morning hours) thoracic oppression, and a cough. In this entity, there is the participation of the cells as well as of inflammatory mediators that together develop its pathology and cause the hyperreactivity of the smooth muscles of the airway, culminating in its obstruction and the restriction of airflow, avoiding the achievement of adequate hematosis (that is, a correct gaseous exchange of O_2_ and CO_2_ in the blood) [3]. According to the latest GINA 2020 update, the data suggest that asthma affects approximately 300 million persons worldwide. Therefore, it is considered a public health problem that generates high management costs for the health systems of different governments [4]. In Mexico, it is estimated that 7.0% of the population suffers from asthma, that is, 8.5 million inhabitants, according to the World Health Organization (WHO) database. With respect to mortality, in 2011, the Mexican National Institute of Geography and Statistics (INEGI) showed a statistical datum of 291 deaths per 100,000 inhabitants [5,6]. It was confirmed that this entity similarly affects the different socioeconomic strata, being distributed worldwide in all geographic regions. In the last 30 years, its prevalence and incidence have increased in highly industrialized countries, a situation probably related to the factor of environmental contamination. Asthma can appear at any age, and its prevalence is observed in the masculine gender in boys aged less than 10 years. Later, during puberty, its prevalence levels off in both sexes, while at adult age, its incidence focuses preferentially on the feminine gender [7]. It has been suggested that this prevalence in the female gender is associated with hormonal fluctuations during menstruation, pregnancy and menopause. Animal studies using genetic deletions of estrogen receptors have shown that estrogen signaling promotes allergen-mediated type 2 airway inflammation. In addition, ovarian hormones have been shown to be important for interleukin 17 A- mediated airway inflammation [8].

## 2. Overview of Asthmatic Pathophysiology

Diverse studies and/or publications have fully described the physiology of asthma. In general, it is considered a chronic disease of the airways whose main anatomical–functional alterations comprise the following:(a)Airflow obstruction: In the airway of a patient with asthma, we find the generation of hypertrophy and cellular hyperplasia, giving rise to an increase in the bronchial smooth muscle mass. Generally, this increase is induced by the fibroblasts and pericytes (also denominated Rouget cells) present in the vascular endothelium, which possess the capacity to convert into muscle cells through a process of differentiation. A bronchospasm (the sudden response of an individual with asthma) generates the contraction of the bronchial smooth muscle on being confronted with diverse stimuli, causing the narrowing of the airway with the diminution of the flow. It is known that different factors can regulate bronchial smooth muscle tone, highlighting the epithelial and endothelial cells, mastocytes, and macrophages, as well as inflammatory cells (eosinophils, lymphocytes, neutrocytes, and basophils), which release proinflammatory substances (such as histamine, eicosanoids, and platelet activating factor (PAF)). In addition, direct stimuli release acetylcholine, which induces the bronchospasm to a greater degree [2,9].(b)Nonspecific bronchial hyperreactivity (NBH): NBH in an asthmatic is an exaggerated response to a stimulus (contaminant, allergen, exercise) that induces a more intense bronchospasm than that of a normal individual [3,9].(c)Inflammation and remodeling of the airway: Inflammation, the principal contributor to the expression of asthma, generates an increase in the reactivity of the airway and recurrent episodes of wheezing, respiratory difficulty, cough, and thoracic oppression. Generally, this inflammation produces edema, angiogenesis with dilation and congestion, and smooth muscle hypertrophy and hyperplasia. Thus, the increase and size of the vessels contributes to the thickening of the bronchial wall, favoring the limitation of the airflow and generating bronchoconstriction. There is evidence that the inflammatory process produces an alteration in the respiratory epithelium. The extension of this alteration or damage may be attributed to a dysfunction in the epidermal growth factor receptors (EGFR), which regulate the epidermal growth factor (EGF), which are indispensable for normal and adequate re-epithelization. The EGF stimulates epithelial proliferation and the production of matrix metalloproteases (MMP), which degrade the extracellular matrix (ECM) and maintain an equilibrium with transforming growth factor beta (TGF-β), which increases the synthesis of the ECM components and inhibits the production of MMP [3,9,10,11,12,13].

Airway remodeling is characterized by the thickening of the reticular lamina (generally with deposits of subepithelial and perivascular fibrin, hyperplasia, the mucosal glands, and vascular and smooth muscle). The fibroblast possesses a relevant function during this process, due to the fact that it produces a large number of cytokines, of growth factors, and that it induces the synthesis of hyaluronic acid and other ECM proteoglycans. In addition, fibroblasts activate compensatory mechanisms of bronchial inflammatory damage (epithelial regeneration and the deposit of collagen between the muscle layers) induction of angiogenesis, vascular infiltration, and vasodilation. Taken together, all of these changes tend to be the result of direct damage to the epithelium to produce remodeling [3,9,10,11,12,13].

It is noteworthy that atopia, also called allergy (current concept), is the most identifiable factor of asthmatic crises, due to the fact that it is genetically predisposed to producing Immunoglobin E (IgE) as an antigen-specific response to allergens (such as dust, animal epithelia, pollen, or synthetic fibers), which are commonly innocuous for the majority of non-asthmatic individuals. In general, the production of IgE depends on the B lymphocytes, and it is regulated by Interleukin 4 (IL-4) and interferon gamma (IFN-γ), synthetized by the TH2 lymphocytes (type 2 cooperator T lymphocytes) and the TH1 lymphocytes (type 1 cooperator T lymphocytes), respectively. Thus, when an individual is sensitive to an allergen, the latter adheres to their IgE and in turn, this interleukin binds to the mastocyte’s cellular membrane to release inflammatory mediators that favor the clinical manifestations of asthma. The synthesis of IgE initiates, after repeated exposure to an allergen, for it to be carried to the lymph nodes, which is where the immunologic memory is imprinted [3,14,15].

## 3. Risk Factors for Asthma

It has been established that, in order for diseases to occur, it is necessary for them to coincide with different elements. Asthma, as a multifactorial disease, conforms to the incidence of different factors of the patient’s macro- and microenvironment. The risk factors and triggers of this disease are multiple, the most relevant of these being genetic, infectious (viral, bacterial, fungal, and parasitic), occupational and environmental (aeroallergens, dust, pollen, suspended particles, chemical irritants, tobacco), related to climate changes, dietetic, and obesity-related. In addition, it is thought that the severity of this entity is influenced by age, sex, pregnancy, immaturity of the immunological system, and the atopic march. Likewise, the interaction of the previously mentioned phenomena (obstruction, NBH, inflammation, and airway remodeling) determines its clinical responses and treatment responses [3,16].

The association of asthma with the atopic march or allergies: At present, this is considered to be factor number one for developing asthma. The respiratory system, the skin, the mucosa and the digestive tract participate in this association, as a consequence of a complex immunological disorder. The factors that exert an influence on the atopic course are hereditary in type, related to intrauterine sensitivity and/or maternal immunity, as well as environmental and infectious factors related to habits and/or lifestyle. It tends to be characterized by a course denominated the “clinical triad”, in which atopical dermatitis, allergic rhinitis, and/or the induction of asthma are present. In general, it is observed that atopical dermatitis presents between birth and 6 months of age. Later, gastrointestinal disorders appear, mainly during the second year of life. Between 3 and 7 years of age, there can be the initiation of disorders in the upper respiratory tract, culminating in the establishment of asthmatic crises between the ages of 7 and 15 years. To speak of the “atopic march” is to refer to the allergic progression mediated by IgE, which begins with atopic dermatitis and a food allergy in infancy, followed by aeroallergen sensitization at preschool age [8,16,17].

Infectious antecedents: The respiratory epithelium separates the external environment of the internal pulmonary medium, controlling the inter- and transcellular permeability of the passage of pathogens and access to the antigen-presenting cells involved in the immune inflammatory response. Infections produced by bacteria, viruses, fungi, or parasites activate different cellular responses and/or signaling pathways that generate changes on the cellular surface and modify the response to stimuli and/or the previously mentioned infections. Diverse receptors can act during the induction of asthma, the most representative being pattern recognition receptors (PRRs), protease-activated receptors (PARs), and those denominated Toll-like receptors (TLRs), which together favor the identification of microbe-specific molecules and pathogen-associated molecular patterns (PAMPs) [16,18].

Viral infections are an important cause of asthmatic exacerbation and tend to be a causal factor for the development of childhood asthma. It is considered that the first white cells that can be infected by rhinoviruses (RV) are the epithelial cells (between approximately 45 and 50% of cases), while the syncytial respiratory virus (SRV) can initiate an infection in 21% of cases. Both viruses are significantly associated with a high probability of inducing asthma and producing early exacerbations, especially in individuals younger than 2 years of age [16,19,20].

In the case of fungal infections, the fungus-synthesized proteases that tend to grow in humid places can comprise an important factor in producing and increasing the incidence of rhinitis, asthma, and other respiratory diseases. In general, these proteases are related to an increase in the granulomatous response and/or in the production of immunoglobins (principally E and G) [16,21].

With respect to microbial infections, it is convenient to consider the so-called “hygiene theory”, which suggests that the lack of exposure to bacteria and/or endotoxins favors the persistence of the TH2 response, increasing the possibility of presenting an atopic disease. Contrariwise, it is suggested that exposure at a young age to bacterial products can prevent subsequent allergic sensitization; therefore, the asthma being directed toward the TH1 pathway; that is, its differentiation is directed toward Interleukin 12 (IL-12) and IFN-γ. Such is the case of the use of lactobacilli in allergic and non-allergic children, where these lactobacilli have been proposed to possess a potential benefit, especially in the treatment of atopic dermatitis. In this context, it is known that probiotics are potent inducers of IL-12, stimulating TH1 immunity and, together the natural maturation process of the digestive immunological barrier, has allowed the digestive tract to be more resistant to the distinct bacterial aggressions, reducing the presence and/or the incidence of asthma [16,22,23].

Dietetic factors and obesity: Epidemiologic data suggest that a poor diet, together with obesity (individuals with a body mass index (BMI) above 30 kg/m^2^) can increase the risk of suffering from asthma. In the specific case of a diet high in saturated fats and deficient in polyunsaturated omega-3 fats, fiber, vitamins (especially A, C, and D), magnesium, and selenium, this has been related to the inflammatory induction of the respiratory tract and a direct association with the presence and/or incidence of asthma [17,24,25]. A study conducted by Maciag and Phipatanakul (2019) indicated that pregnant women treated with high doses of 25-hydroxyvitamin D (Calcidiol (2400 and 4000 IU daily)) can reduce by 25% the risk of their offspring having asthma. The latter suggests a balanced diet, one with adequate concentrations of Calcidiol, induces beneficial effects in the uterus and diminishes the possibility of developing asthma [17]. Arias-López et al. (2018) consider that, over the past 30 years, vitamin D levels have diminished in the population in general (particularly in pediatric individuals). Such a reduction suggests that it is related to an increase in some diseases, such as asthma [25].

On the other hand, it has been observed that obesity can exert different negative effects on the lungs, highlighting a reduction in pulmonary function (maximal expiratory volume at the end of the first second (FEV1)/forced vital capacity (FVC)), as well as an increase in the appearance of dyspnea and wheezing. Thus, it has been suggested that obese non-asthmatic patients manifest cardiorespiratory symptoms similar to those of individuals with asthma, principally brought about by a narrowing of the airway due to the accumulation of fat in the thorax. Additionally, obesity can generate a state of systemic inflammation (due to the high concentration of adipocytokines, such as leptin, resistin, an inhibitor of the activation of the plasminogen, tumor necrosis factor alpha (TNF-α), IL-6, and angiotensinogen), which, to a certain degree, acts on the lungs, precipitating the initiation of asthma. Likewise, epidemiologic evidence indicates that patients with asthma who are obese or overweight experience a higher number of emergency room hospitalizations in comparison with asthmatic non-obese individuals. Asthmatic persons with obesity are also considered to have a reduced response to glucocorticoids. Thus, they need higher doses of the latter to improve the control of their asthmatic condition, a situation that, in the long term, can give rise to more significant secondary effects [16,26,27].

Genetic factors: The majority of scientific evidence indicates that the genes play a determining role in asthma. Its heritability ranges from 36–79%, without the existence of a well-defined pattern. Studies have been conducted in various chromosomal regions that contribute to the susceptibility of inducing asthma, although these principally address chromosome 5q31–33,35, which maintains a relation with TH2 and Interleukins 4, 5, 9, and 13. However, despite the presence of these polymorphisms, the participation of other triggers is necessary to determine the type, severity, prognosis, and treatment of this pathology. To date, there are more than 100 genes reported in association with asthma or related to their phenotypes. Table 1 shows the main chromosomes related to this entity [9,16].

Although asthma affects all ethnic groups, its incidence is more frequent in some economically disadvantaged groups. In addition to genetic factors, some authors consider that socioeconomic status (SES) and access to healthcare generates greater variability in asthmatic prevalence among the different ethnic groups. SES is highly related to ethnicity and it is a risk factor for asthmatic morbidity, especially in poorly developed countries where environmental conditions, stress and psychological/cultural factors play an important role in its incidence [28,29,30].

Occupational, environmental, and pharmacological agents: Asthma of occupational and environmental origin entertains a close relationship to being caused by exposure to different aeroallergens. Both types of asthma can favor the expression of genes of hypersensitivity, as well as the exacerbation and/or presentation of their symptoms. There are more than 400 aeroallergenic agents, among which are dust (mainly of wood), pollen grains, latex proteins, animal urine and hair, dandruff, mite and fungal proteases, suspended particles [carbon monoxide (CO), nitrogen dioxide (NO_2_), sulfur dioxide (SO_2_), ozone (O_3_) and diesel particles], chemical irritants (acid anhydrides, polyisocyanate polymers, and platinum and persulfate salts), and tobacco smoke, in addition to climate changes, and stress itself. (Table 2) [2,16,22,31,32,33,34,35,36].

It is known that exposure to some aeroallergens exacerbates asthma and increases the risk of acute crises in patients with allergy, mainly in children. Therefore, it is important to reduce and to avoid these in order to improve quality of life, thus diminishing the need for employing drugs [2,22]. Relevant data in this context are the following: (a) the exposure of infants to the Great Smog of 1952 (London, UK), in which the rate of this entity increased by approximately 20% [37], and (b) Maciag and Phipatanakul (2019) observed that children who were not breast-fed have a greater incidence of this disease in comparison with those who were breast-fed. Additionally, the authors suggest that vitamin D confers protection against the development of asthma, especially against exposure to the aeroallergens in the traffic-related air contamination of industrialized countries [17].

Antibiotics and non-steroidal anti-inflammatory drugs (NSAIDs) are found among the main drugs that trigger asthmatic crises. In the first case, the β-lactams (such as penicillin, ampicillin, and amoxicillin) have been related to certain allergies and there is evidence that, on their being administered during the first years of an infant’s life, can alter the development of intestinal microbiota, generating a greater risk of having asthma [34]. With regard to NSAIDs, practically all of these are potentially capable of precipitating asthma, this due to their capacity to diminish prostaglandins (PGs) and to increase cysteinyl leukotrienes (CysLT) on inhibiting cyclooxygenase (Table 2) [38,39,40]. Ishitsuka et al., (2020) conducted a review in order to document whether Acetaminophen (N-acetyl-p-aminophenol) presented adverse effects similar to those of NSAIDs. The authors’ results indicated that, in high-risk patients such as the elderly, children, and pregnant women, that it can be a drug that cause kidney dysfunction, gastrointestinal injury, asthma and/or bronchospasms [41].

## 4. Asthma and Its Relationship with Oxidative Stress

As already mentioned, asthma is a chronic inflammatory disease of the respiratory tract whose etiology is multifactorial. Over the past years, the scientific evidence has increased of oxidative stress (reactive oxygen species (ROS), as well as reactive nitrogen species (RNS)) on the induction, activation, and possible treatment of asthma.

Although the generation of both types of free radicals (RONS) form parts of different physiological responses of the organism, on occasion, when an asthmatic crisis presents as a response to diverse stimuli (among these are environmental and occupational allergens), their levels can increase, overcoming the antioxidant mechanisms of the airways and causing structural damage and metabolic alterations that together can favor the pathology of this disease [42,43].

Diverse investigations have produced evidence of and have demonstrated an oxidative process that is generated in the respiratory pathways, a process that is very complex, which it would be difficult to fully describe in this manuscript. Generally, RONS can be produced by different cells, under normal physiological conditions as well as during an asthmatic crisis. In the specific case of ROS, the superoxide anion (O_2_^•−^) is importantly generated by the mitochondria and, due to the fact that it is considered a potential microbiocidal agent, its most relevant role is focused on the neutrophils, eosinophils, monocytes, and macrophages. During the respiratory burst of the leukocytes, O_2_^•−^ is generated through the activation of the enzyme NADPH oxidase. Later, it is transformed into hydrogen peroxide (H_2_O_2_) by the action of the Superoxide dismutase enzyme (SOD). This new radical increases its oxidative potential by means of peroxidase enzyme of eosinophils (EPO) and the myeloperoxidase (MPO) of the neutrophils in order to act as a microbiocide. There is evidence that both radicals (superoxide anion and hydrogen peroxide), together with hydroxyl radical (HO^•^), plus the RNS [nitric oxide and (NO^•^) and peroxynitrite (ONOO−)], can produce inflammatory effects and oxidation phenomena in the proteins of the respiratory pathway in patients with asthma [42,43,44]. On the other hand, NO^•^ is considered a relatively stable radical whose important functions in the lungs comprise regulating the pulmonary vascular tone, stimulating the secretion of mucin, modulating mucociliary clearance, and exercising bactericidal action. There are three nitric oxide synthase isoenzymes in the lungs (iNOS1, iNOS2 and iNOS3), whose main function is to catalyze the conversion of L-arginine into NO^•^ and L-citrulline. iNOS1 and iNOS2 are always present in the organism at a constant concentration, while iNOS3 is considered inducible when faced of diverse stimuli. iNOS2 is a strong source of NO^•^ in the healthy lung, and there is evidence that certain abnormalities in its genotype and expression favor the increase in this radical and its relation with the incidence of asthma. In addition, it has been suggested that persons with asthma can have three times more NO^•^ in the lower airways and in exhaled air. Therefore, excessive synthesis is related to significant inflammatory processes. Another possible mechanism by which RNS favor asthma is through the processes of nitrosation (a covalent addition between RNS and SH and/or amine groups) and nitration (RNS with aromatic rings) of proteins, which are related to the altered activities of signaling enzymes and molecules [42,43,44,45,46].

In summary, it has been observed that the increase in RONS levels is directly related to the severity of asthma. Higher RONS levels induce respiratory tract inflammation upon activating different transcription factors, highlighting the Nuclear Factor Kappa-Light-Chain-Enhancer of Activated B Cells (NF-κB), mitogen-activated protein kinase (MAPK), and activator protein-1 [43,47,48,49]. These transcription factors promote the expression of IL-6, IL-8, and TNF-α, inducing an activation of the inflammatory cells that damages and affects the lung tissue [40,50,51].

## 5. Classification and Diagnosis of Asthma

Initially, this entity was classified considering its causes, intensity, and airway obstruction (frequently estimated by means of maximal expiratory volume at the end of the first second (FEV1) and/or peak expiratory flow (PEF)), a classification that was inappropriate due to the fact that asthma is a disease of multifactorial etiology. Currently, chronic asthma is classified according to GINA 2020 guidelines on controlled, partially controlled, and non-controlled asthma. For investigative purposes, asthma continues to be classified as intermittent, slightly persistent, moderately persistent, and severely persistent (Figure 1). It is noteworthy that these classifications are considered after carrying out a diagnosis, which permits the determining of the scheduling of its pharmacological therapy. In general, the following parameters are taken into account:(a)Symptom control: good control, partial control, out of control.(b)Future risk: depends on FEV1 and other factors that increase the risk of exacerbations, irreversible obstruction, or drug-associated adverse effects.(c)Severity (established based on the clinical history, especially considering the level of medication employed to maintain symptom control): intermittent or slight, moderate, and severe [3,4,5,52].

The diagnosis is based on an analysis of the clinical history of the patients in which the social and environmental milieu, the familial antecedents, and a physical examination are taken into consideration. In addition, the symptoms of respiratory obstruction (cough, wheezing, thoracic oppression and, in severe cases, respiratory difficulty) are evaluated. For full confirmation of the disease, pulmonary lung function tests are employed.

According to the asthmatic diagnosis algorithm (Figure 1), tests start with a spirometric evaluation to determine the Tiffeneau index (the relationship between FEV1 and forced vital capacity (FVC)), which indicates a possible obstructive pattern when its value is below 70%. Subsequently, the bronchial asthmatic reactivity test is performed by administering a short-acting beta agonist to verify obstruction reversibility and differentiate with chronic obstructive pulmonary disease (COPD).

If the result is normal, two additional tests should be performed with the purpose of discarding the diagnosis of asthma. The primary determination of the variability of the maximal expiratory flow (MEF) is obtained by conducting serial measurements throughout the day and employing the following formula:MEF = maximal MEF − minimal MEF × 100/maximal MEF

A value of ≥20% over 3 days, across 2 weeks, is highly suggestive of asthma.

The second test comprises the stimulation of a bronchospasm induced pharmacologically with methacholine (cholinergic agonist). If the patient presents a bronchospasm at low doses of the drug, the FEV1 will diminish by 20%; this translates into a positive result and confirms the diagnosis of asthma [3,4,5,52].

## 6. Management, Control and the Main Pharmacological Treatments of Asthma

Throughout history, different pharmacological schemas have been used to control and reduce asthma symptoms. The first therapies were unorthodox and absurd since they included “pouring a gallon of ice water on the patient’s back from a height of three meters”, “applying electric current to the vagus nerve, the accessory, and the sympathetic through an electrode placed on the neck or inserted into the patient’s nose”. Even smoking was considered as a therapy to avoid bronchospasms. The first antecedent of the 20th century was found in the Orville Brown treatise, where controlling the disease through diet, massages with chest vibrations, breathing exercises and avoiding exposure to allergens, were recommended. Essentially, the pharmacological era of asthma started with Dr. Hirsh (1937) when he prescribed theophylline (methylxanthine alkaloid) suppositories; however, due to their extensive adverse effects, they are no longer used. Later, inhaled treatments were introduced, mainly belladonna, organic nitrites, estramonium, and atropine [53].

During the 1950s, the β_2_ adrenergic agonist agents (β2AA) such as albuterol (also called salbutamol) and terbutaline began to be distributed and were considered drugs of first choice at that time. Some years later, the combination of β2AA and corticosteroids was explored to reduce both bronchospasms and inflammation. Corticosteroids were initially prescribed systemically; but they were discontinued due to their high adverse effects. It was not until 1970 that betamethasone and beclomethasone (long-term inhalable corticosteroids (ID)) began to be used and, practically, at the end of the 20th century, the treatment of asthma left its empirical nature behind, and the proper management of asthmatic treatment started under the consensus of national and international guides [53,54].

Currently, the control of chronic asthma seeks five primary goals (Table 3), so the medical guidelines propose a therapeutic scheme in stages; that is, considering that it is a chronic inflammatory entity, the basis of treatment should be early anti-inflammatory therapy, rather than symptomatic treatment. This essentially involves avoiding risk factors to establish a suitable treatment, classifying it by its intensity, increasing the number, frequency and dosage of drugs until possible remission is reached, then carefully reducing the drug dose until it is as small as possible to retain remission; finally, the treatment must be individualized and modified to maintain correct control of the symptoms. In summary, the therapeutic strategy should be divided into: (a) environmental control, (b) patient education, and (c) adequate pharmacological management and immunotherapy [3,52].

The Global Initiative for Asthma (GINA) is the most utilized set of guidelines and, in its most recent update (2020), it assembles data derived from the Spanish Guidelines on the Management of Asthma (GEMA 2018) and from the Mexican Asthma Guidelines (GUIMA 2017). This information brings together evidence on the factors associated with asthma, its diagnosis and management in its different stages, and suggestions about the new therapeutic options available in recent decades (especially information related to monoclonal antibodies) [4]. Figure 1 shows the steps of the treatment suggested by GINA 2020.

Clinical efficacy largely depends on adherence to treatment. In general, the drugs used in the management of asthma are divided into two groups, those with bronchodilator potential that achieve symptomatic improvement by relaxing the smooth muscle of the airway (β2AA, anticholinergics, and methylxanthines) and, on the other hand, inflammation-controlling agents (ID and antileukotrienes). Table 4 shows the main characteristics of the drugs used in asthma therapy [2,3,4,5,6].

## 7. Alternative Therapies for Asthma Control

As previously stated, drugs seek to reduce the inflammation of the respiratory tract and alleviate bronchospasms. Unfortunately, symptoms may reappear when administration is discontinued. Despite its existence and the fact that the administration is based on international and national guidelines, controlling asthma continues to be a great challenge. Some studies indicate that more than 50% of asthmatic patients are uncontrolled, even with maintenance treatments (ID combined with a long-acting β2AA). Therefore, new alternatives should be considered:

Immunomodulatory agents: In the past decade, notable advances have been made in biological therapy with monoclonal antibodies (MoAb). Omalizumab was the first and is the only MoAb approved for asthmatic individuals older than 12 years, administered subcutaneously on a monthly basis. Its mechanism of action is based on the protein–protein interaction between its high-affinity receptor and IgE, which prevents binding to mast cells and basophils, decreasing the release of inflammatory mediators [3,55,56]. Due to its interaction with IgE, Omalizumab has shown an important role in atopic or allergic asthma. However, considering that asthma of non-allergic origin presents a similar inflammatory process (increase in TH2 lymphocytes, activation of mast cells and infiltration of eosinophils), this has opened the field of study to other possible MoAbs. Table 5 summarizes the immunomodulatory agents under development and their main characteristics [55,56].

Herbal and/or antioxidant strategies: The WHO estimates that more than 80% of the Earth’s inhabitants to trust and have used Traditional Medicine/Complementary and Alternative Medicine (TCAM) for their primary healthcare needs. Among the objectives of this concept are to treat, diagnose and prevent certain diseases such as obesity, diabetes, hypertension, and cancer. Chronic respiratory diseases such as asthma, have also been included [57,58]. Dealing with herbal medicine or herbal interventions for chronic asthma would be a very broad topic difficult to fully address in this review; however, it is possible to highlight their benefits as adjuvant supplementary therapies to the main pharmacological treatment and their possibilities to reduce its doses and potential adverse effects of the latter. In general, this type of intervention is organized into:(a)Diet, Vitamins and Food Supplements

Various studies have confirmed that food and nutrients can protect the airway from oxidative damage through different mechanisms. For example, some vitamins (soluble (vitamin C) and fat-soluble (vitamin E)) are considered an important defense against RONS. Similarly, carotenoids (α and β), vitamin A, and lycopene have shown significant potential antioxidant effects. Those results also show that asthmatic individuals present lower concentrations of vitamins A, C and E, which may enhance their symptoms. The same happens if selenium levels are low, since this element is essential for the enzyme glutathione peroxidase (GPx) to function properly and to reduce the amount of H_2_O_2_, preventing the lipid peroxidation of the cell membrane [43,59,60].

Some evidence shows that asthmatic adults with a diet low in antioxidants may have a low FEV1, FVC and more frequent exacerbations. These alterations tend to balance when they ingest a nutritional supplement enriched with antioxidants [43,61]. Research carried out by Hernández et al. (2013) and Peh et al. (2015) with the vitamin E γ-tocotrienol isoform showed that this isoform can inhibit oxidative damage by promoting the production of endogenous antioxidants in the lungs. The process was related to its potential to increase Nrf2 levels by blocking Nuclear Factor Kappa-Light-Chain-Enhancer of Activated B Cells (NF-κB). They also noted that the airway hyperreactivity improved and the lipopolysaccharide-induced neutrophil infiltration decreased [62,63]. All these vitamins and nutrients are essential for humans and are found in different fruits, vegetables, seeds, cereals, seed oils, nuts, shellfish and some red meat [43,59]. Another known and relevant therapy is thiol antioxidants, which induce the conversion of glutathione. N-acetyl cysteine (NAC). NAC treatment is known to reduce the need of bronchodilators and the responsiveness of the airway when irritated by inhaling diesel exhaust. Furthermore, it has also been linked to its potential to significantly inhibit RONS and lipid peroxidation [43,64].

(b)Plants and Natural Extracts

The consumption of medicinal plants to treat different diseases has increased worldwide in recent years. Many asthmatics use plants alone or in combination with prescribed medications to try to reduce and/or control symptoms. Two randomized meta-analysis studies collected different controlled trials to evaluate the effectiveness of herbal medicine in adults with this condition. In the first study (2010), 26 trials on approximately 20 herbal preparations considered in Chinese, Indian and Japanese therapies were analyzed; the evaluation parameters were divided into primary (lung function, number of exacerbations and reduction in the use of corticosteroids) and secondary (symptoms and adverse effects). In summary, their results showed very few relevant data on reductions in exacerbation. Six studies with Chinese therapy indicated changes in FEV1 and a study where 1.8-Cineol (eucalyptol) was used showed evidence of a reduction in the daily dose of oral steroids [65,66]. The second study (2016) consisted of 29 trials that included 3000 participants with an average age of 43 years. Mainly, the anti-asthmatic effect of the plants *Glycyrrhiza uralensis* (licorice root), *Angelica Sinensis, Pinellia ternata,* and *Astragalus membraneceus* (the latter two included in traditional Chinese medicine) were evaluated. The diagnosis and treatment of the disease was performed on the basis of parameters of the GINA. The analysis confirmed that almost all plants, as a complement to routine therapy, improved the asthmatic control and lung function. Likewise, the frequency of acute exacerbations and the use of salbutamol decreased [66]. In addition to the examples mentioned above, there are records on the plants “Perpetual” (*Helichrysum stoechas*), “Eucalyptus” (*Eucalyptus globulus*), “Rosemary” (*Rosmarinus officinalis*), “Ginger” (*Zingiber officinale*) and “Elecampane” (*Inula helenium*) that due to their aromas and relaxing effects, they can reduce asthma symptoms [43,67]. It is worth mentioning that the traditional consumption of these medicinal herbs is, in general, orally, on a weekly basis and for one month (mainly as aqueous extracts or tea). According to each area or geographic entity where they are used, their preparation includes the use of the plant alone or a mixture derived from the leaves, stems, roots and/or fruits.

On the other hand, garlic (*Allium sativum* L.), a bulbous perennial plant with a peculiar purgative flavor, has shown antimicrobial, antifungal, analgesic, antihypertensive, anticancer, antioxidant and anti-asthmatic capabilities, attributed to different phytochemicals in its chemical composition, among which organic sulfur compounds (such as diallyl sulfide (DS)) stand out. In the case of its anti-asthmatic potential, studies conducted with DS showed that when it acts on Nrf2 it can reduce ovalbumin-induced infiltrated inflammatory cells and proinflammatory cytokines in BALF mice [57,68].

Finally, research carried out with resveratrol, a phenolic compound found in grapes, indicated that it can decrease ROS production and reverse the high levels of TNF-α and iNOS in the lungs of obese C57BL/6 male mice sensitized with ovalbumin (OVA). This protective effect is probably related to its ability to regulate the decrease in the phosphoinositide 3-kinase-protein kinase B pathway, while regulating or producing an elevation of inositol polyphosphate 4 phosphatase [69,70].

(c)Therapy and Integrative Medicine (IM) for Asthmatic Control

Due to the multifactorial nature of asthma and the fact that the disease has neuromuscular (bronchospasm), immunological (inflammation) and psychological components, another possible alternative for its control and treatment can be through an integrative medicine (IM) approach [this concept was defined by the National Center for Complementary and Integrative Health—NIH (NIH-NCCAM), in other words, approaching all the pathways of its pathophysiology through an integrative medicine that combines the use of complementary evidence-based therapies with conventional medicine. As mentioned already, some studies suggest that more than 50% of asthmatic individuals (children and/or adults) express dissatisfaction with conventional control treatments (β2AA, anticholinergics, methylxanthines, ID and antileukotrienes) and/or unconformity due to the resultant side effects. Among the most common and used IMs are dietary and nutritional therapies, herbal remedies, homeopathy, acupuncture, massage, yoga breathing exercises, relaxation and mind–body therapies (MBSR), and Qigong (a traditional Chinese medicine combining movement, meditation and breathing techniques) [71,72,73].

Since Arthur Kleinman [74] suggested to the medical field that individuals may have more options to treat their diseases, not just the conventional biomedical approach, research evaluating the combination of traditional practices with conventional therapies have increased. Those that reduce stress and anxiety favor a better attitude towards the disease and therefore asthmatic attacks can also be reduced. One of the first studies (1986) was a prospective, randomized, single-blind controlled trial with 44 adults of both genders. In that study, the use of bronchodilators was reduced and a 70% improvement in their induced airway hyperresponsiveness was evidenced by methacholine after 6-week hypnotherapy [75]. Hypnotherapy has also been successful in the pediatric population; Anbar (2002) confirmed an 80% decrease in asthmatic symptoms (cough, shortness of breath, chest pain and hyperventilation) in children treated with this therapy. This type of therapy has also contributed to reducing school absenteeism [76].

Other studies where different types of yoga programs were analyzed as a treatment therapy showed a progressive improvement in lung function in both children and adults during the course of the treatment. However, the results on the benefits of this type of therapy in asthmatic children were not completely significant and definitive. Thus, is necessary to explore this technique in new clinical trials [77,78,79].

Mindfulness-based stress reduction (MBSR) is another mind–body approach with some promise of success. This technique has been frequently used for the treatment of anxiety and pain as well as symptoms related to cancer. Pbert et al. (2012) compared a randomized controlled trial of an 8-week MBSR program against an educational program in adult patients with asthma. At the end of the evaluation period, they confirmed clinically significant improvements in their quality of life and stress reduction, which had a favorable impact on their lung function [80].

Finally, research carried out by Cotton et al. (2011) and Shen and Oraka (2012) indicated that approximately 70% of children and adolescents, belonging to poor socioeconomic backgrounds, with poorly controlled asthma, used complementary and alternative medicine for the management of their disease. The most used therapies were relaxation (85%), breathing techniques (58%), and herbal products (12%). The results of both studies conclude that these therapies may be useful to improve asthmatic control [81,82].

## 8. Perspectives and Conclusions

This review demonstrates the attention that different health professionals have shown to understand more clearly and concisely the pathophysiology of asthma. The data presented confirm that, over the years, the knowledge about the mechanisms involved in the development of this entity, as well as its classification, treatment and control have increased and evolved. The GINA compiled the most outstanding information from GEMA and GUIMA to create the most current version of a catalog or guide for its diagnosis, management and treatment that supports doctors, researchers and scientists.

Currently, GINA 2020 suggests classifying asthma in intermittent cases—slightly persistent, moderately persistent and severely persistent—which allows us to define appropriate guidelines for drug therapy. Given that asthma cannot be completely cured, it is advisable to seek suitable control to decrease severity and possible risks in patients. It is also convenient to encourage people to know more about their disease (patient education) and to reduce or avoid risk factors (occupational/environmental situation), since this is where various allergen agents that activate symptoms are located.

It is of utmost importance that health professionals make an adequate diagnosis based on the patient’s medical record considering their social, environmental and family history. When performing a physical examination, evaluating the symptoms of respiratory obstruction (cough, wheezing, chest tightness and possible respiratory distress), is also relevant to fully confirm the disease after analyzing lung function tests established by the asthmatic diagnosis algorithm (Figure 1).

Regarding therapeutic management, which has ceased to have an empirical nature, evolving through the consensus of national and international guidelines, it is convenient to remember that this chronic inflammatory disease should preferably be treated early with an anti-inflammatory therapy. The information shown in this article classifies the drugs used in asthmatic management into two groups, those with bronchodilator potential that achieve symptomatic improvement by relaxing the smooth muscle of the airway (β2AA, anticholinergics and methylxanthines) and, on the other hand, the agents that control inflammation (ID and antileukotrienes).

According to the latest GINA 2020 version, this entity should be managed through five steps. In the first, a combination of budesonide and formoterol should be used at low doses, eliminating monotherapy with β2AA (such as albuterol). During the next step, it would be advisable to start using an ID on a daily basis and include a short-acting β2AA on demand. Subsequently, three possibilities are considered: a) maintaining the IDs combined with long-acting β2AA, b) IDs at low doses plus an antileukotriene, or c) start using a medium-dose of IDs. In the fourth step, two options are considered: a combination of ID (medium dose) and a long-acting β2AA or ID and the same dose plus an antileukotriene. Finally, during the fifth step, it is suggested that a pulmonologist, allergologist or immunologist assesses the use of high or medium doses of ID combined with long-acting β2AA.

However, as with many drugs, IDs can generate different adverse effects when accumulated (mainly in oral and pharyngeal mucous membranes) or used at high doses. These undesirable effects are mainly related to the suppression of the hypothalamic–pituitary–adrenal glands axis and the probability of inducing osteoporosis, cataracts, skin atrophy, weight gain, diabetes, hypertension, psychological disorders and immunosuppression. Unfortunately, if patients interrupt their drug therapy, their symptoms may reappear or increase. Thus, asthmatic control continues to be a great challenge, given that more than 50% of patients are not completely controlled.

Asthmatic treatments continue to evolve in order to reduce symptoms and improve the quality of life of patients. In recent years, notable advances have been made in the biological therapy of monoclonal antibodies (MoAb), among which omalizumab is the only MoAb approved for asthmatic individuals over 12 years of age, administered subcutaneously on a monthly basis and with a mechanism of action based on the protein–protein interaction between its high-affinity receptor and IgE, avoiding binding to mast cells and basophils to decrease the release of inflammatory mediators.

On the other hand, considering that asthma triggers inflammatory processes and that the scientific evidence of its relationship with oxidative stress (ROS/RNS) has increased, herbal strategies and the use of antioxidants might be supportive to traditional pharmacological treatments, probably to reduce doses and adverse effects. Different scientific evidence has suggested that a diet including fruits, vegetables, seeds and cereals, the consumption of vitamins A, C, and E and the use of plants and natural extracts (phytotherapy) contribute to reducing and/or controlling asthmatic symptoms. Different plants, such as *Glycyrrhiza uralensis, Angelica Sinensis,*
*Pinellia ternata*, *Astragalus membranaceus,*
*Helichrysum stoechas*, *Eucalyptus globulus*, *Rosmarinus officinalis*, *Zingiber officinale*, *Inula helenium* and *Allium sativum* L., have shown this benefit and opened up the possibility of analyzing different bioactive compounds, such as diallyl sulfide (extracted from garlic) and resveratrol (obtained from grapes).

It is highly convenient to continue these research studies on other plants and their bioactive compounds (both in preclinical and clinical trials) in order to explore their anti-asthmatic potential, establish their doses and administration intervals and analyze their possible toxic effects in the short, medium and long term. This could contribute to obtaining and preparing new pharmaceutical products for asthma therapy as a relevant goal for the coming decades.

In summary, it is important to consider that, despite the medical progress of asthmatic treatment, its prevalence continues to increase, and its treatment is complicated given the multifactorial nature of the disease. Conventional and newer pharmacotherapies along with integrative herbal and mind–body approaches for asthma may help address symptoms and improve quality of life. A general practitioner and/or specialist must be informed and updated on the subject. They should also be willing to employ different strategies that improve clinical results and are beneficial for quality of life in asthmatic patients.

## Figures and Tables

**Figure 1 medicina-56-00438-f001:**
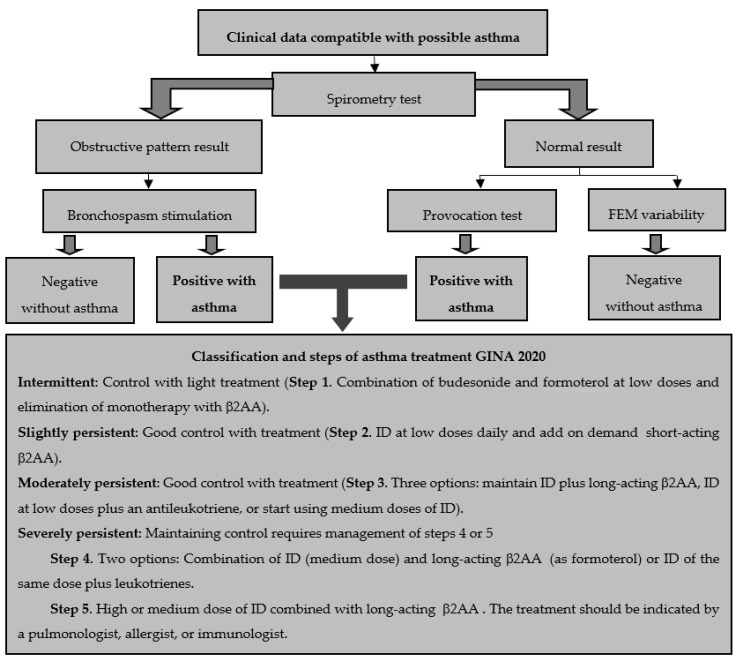
Algorithm of asthmatic diagnosis and steps of its treatment according to GINA 2020.

**Table 1 medicina-56-00438-t001:** Main chromosomes involved in the development and expression of asthma.

Chromosome and Region	Genes, Function and/or Association
1q21.3, 1p31, 1q21 and 1q41	Related to asthma and atopic march. Possible association with filaggrin, PTGR3, FLG, LELP, and TGFβ2
2q32q33	(CTLA-4) associated with cytotoxic T lymphocytes and IgE regulation (ICOS) related to TH2 lymphocytes and cytokine activation (IL-4, IL-5, IL-13)
3q21, 3p25, 3p21.3	(CD80/86, PPARG CX3CR1) associated with asthma and exacerbations. Activation and regulation of leukotrienes and cytokines from TH1/TH2 lymphocytes
4q11–q13, 4q21–23	(CXCL9, CXCL10, CXCL11 SPP1, GSNOR) associated with asthma and rhinitis.
5q31–33,35	Increases production of IgE, eosinophils, cytokines, and interleukins (IL-4, IL-5, IL-9, and IL-13). Relationship with CD14, GRL, GM-FSC and β2AA and/or steroidal receptors
6p 21.3–23, 6q25.1, 6p12	Relationship with the inflammatory process and TNF-α. Also associated with bronchial hyperreactivity (HLA-DRB1, HLA-DQB1, IL-17F)
7p14–p15	(AOAH) susceptibility to asthma and IgE
8q21	(RIP2) associated with severe childhood asthma.
9p21–22	Susceptibility-related type I interferon gene for asthma and atopic march
10q 11.2.10q24	(5-LO, PLAU) associated with asthmatic pathogenesis
11q12–13,11p13	Regulates the beta chain of the receptor for IgE. Additionally associated with anti-inflammatory lung proteins (CC16, CC10. CAT and BDNF).
12q13-24,12q14 and 12q22	Genes related to asthmatic development (early onset and exacerbations)
14q11.2, 14q32.3, 14q24–q31	Genes related to childhood asthma. They are also associated with increased bronchial hyperresponsiveness and decreased response to bronchodilator drugs
16p13.13, 16q24.1, 16p11	(SOCS1) related to adult asthma and atopic march. Possible activation of IL-4 and IL-27.
17q12–q21	(ORMDL3) relationship with early onset of asthma
19q13.1–13.3	(PLAUR) increases synthesis of IgE.
20p31, 20q11.2q13.1, 20q12–q13.2	Possible activation of ADAM 33, MMP9 and CD40 associated with childhood asthma, bronchial hyperactivity and IgE.
21q22.3	(RUNX1) relationship with asthma and IgE
Xp21, Xq13.2–21.1 and CySLTR1	Associated with atopic march and asthma induced by NSAIDs

**Table 2 medicina-56-00438-t002:** Asthma-inducing occupational, environmental, and pharmacological agents.

Agent	Exposure Sites and Individuals	Type of Asthma
Wood dust (cedar, mahogany, ebony, pine and oak)	Carpenters, furniture makers, sawmills	Occupational
Grains (pollen, wheat, barley, coffee, tobacco)	Beekeepers, farmers, bakers, beer industry workers	Occupational
Urine and animal hair	Veterinarians, ranchers, farmers, merchants, daily life (home)	Occupational/environmental
Irritating reagents and/or chemicals (dyes, acid anhydrides, polyisocyanate polymers, and platinum and persulfate salts)	Textile workers, hairdressers, stylists, polyurethane producers, car paint, glue users	Occupational/environmental
Dandruff residue	General employees (offices), daily life (home)	Occupational/environmental
Proteases (mites and fungi)	Field, offices, hospitals, daily life (home)	Occupational/environmental
Suspended particles (CO, NO_2_, SO_2_, O_3_, diesel)	Gas station and refinery employees. Daily life	Occupational/environmental
Tobacco smoke	Employees and workers in general. Daily life	Occupational/environmental
Latex	Health professionals, anyone who uses it	Occupational/environmental
Others (climate changes and stress)	Daily life, employees in general	Occupational/environmental
Medicines (β-lactams and NSAIDs)	Health professionals, daily life (home)	Pharmacological

**Table 3 medicina-56-00438-t003:** Main objectives in the control of chronic asthma.

Number	Objectives
1	Absence and/or decrease in chronic symptoms
2	Reduce the frequency of exacerbations
3	Encourage a normal lifestyle without limitations that allows exercise
4	Maintain normal lung function with minimal adverse effects during treatment
5	Decrease the need to use rescue treatments

**Table 4 medicina-56-00438-t004:** Main drugs used in asthma therapy.

Pharmacological Group	Name (s)	Mechanism of Action	Adverse Effects
**Bronchodilator Potential**			
β_2_ Adrenergic Agonists	First-line drugs. Albuterol, Terbutaline, Pirbuterol and Levalbuterol (Short Action) Formoterol, arformoterol, idacaterol and salmeterol (Long-acting)	They activate adenyl cyclase through the β2AA-receptor and relax smooth muscle, increase mucociliary clearance, and decrease vascular permeability	Uncommon when administered by inhalation. Mainly: Tachycardia, hyperglycemia, hypokalemia, and fine tremors
Anticholinergics	Second-line drugs Ipatropium bromide, and tiotropium bromide. Atropine (prototype agent)	They block the constriction of the smooth muscle of the airways and the secretion of mucus to the muscarinic receptors (M2, M3) of the lung	Atropine: produces thick secretions, blurred vision and cardiac stimulation (all of them limiting its use). Ipatropium bromide: antagonizes both receptors (M3/M2) causing bronchodilation/bronchoconstriction. Effects related to bronchitis, exacerbation of COPD and headaches
Methylxanthines	Third or fourth line drug Theophylline	Ability to relax the bronchial smooth muscle and pulmonary vessels. Its effect is related to the non-selective inhibition of phosphodiesterase.	According to international guides, its use in children has decreased. Short and long-term effects related to nausea, vomiting, arrhythmias, and gastrointestinal bleeding
**Control of inflammation**			
Corticosteroids	Beclomethasone Dipropionate Budesonide Fluticasone propionate Ciclesonide Mometasone	Inhalable agents (sometimes administered systemically) that inhibit the inflammatory response by preventing the release of phospholipase A2 and inflammatory cytokines	High and/or accumulated doses can produce suppression of the hypothalamic–pituitary–adrenal glands axis, osteoporosis, cataracts, skin atrophy, weight gain, diabetes, hypertension, psychological disorders and immunosuppression
Antileukotrienes	Zileuton (a) Zafirlukast, montelukast pobilukast and pranlukast (b)	Two mechanisms of action: (a) Inhibition of the enzyme 5-lipooxygenase and (b) antagonistic effect of the cysteinyl leukotriene-1-receptor (CysLT1)	They can cause headache, rash, insomnia, dizziness, tremor, nausea, vomiting, abdominal pain, heartburn, diarrhea, anorexia, constipation, increased liver enzymes, leukopenia, thrombocytopenia, fever, edema, alopecia, and menstrual irregularities
Chromones	Cromolin (Sodium chromoglycate) Nedocromil	They phosphorylate a myosin-like protein in the cell membrane, responsible for the release of mediators from mast cells and prevent the release of histamine	They appear to have a high safety profile, so they can be used in infants and children under the age of two. In general, they cause irritation in the throat and cough when inhaled. While orally they can cause headache and diarrhea

**Table 5 medicina-56-00438-t005:** Main immunomodulatory agents under development.

Agent	Characteristic and/or Property
**MoAb anti IgE**	
**Quilizumab** **8D6**	8D6 has an affinity for a conformational epitope in the CH3 domain of IgE. Unlike omalizumab, it can bind to low affinity receptors making it more competitive. The other agent has been studied subcutaneously in three doses (150, 300 and 450 mg) in patients with uncontrolled allergic asthma with ID
**MoAb anti IL-5**	
**Mepolizumab** **Reslizumab** **Benralizumab** **TPI-ASM8**	IL-5 is a cytosine modulator of chemotaxis and degranulation of eosinophils. Its receptor is composed of two subunits: (a) specific subunit-α (IL-5Rα) for IL-5 and (b) βc subunit (IL-5Rcβ) responsible for the transduction signal that is shared with specific α subunits of IL-3 receptors and macrophage and granulocyte colony stimulating factor (GM-CSF). Mepolizumab (best known monoclonal antibodies (MoAb) from this group) and together with Reslizumab neutralize IL-5. While Benralizumab acts on IL-5Rα and TPI-ASM8 on IL-5Rβc
**IL-4 Antagonists**	
**Pascolizumab** **Altrakincept** **Pitrakinra** **Dulipumab**	Both IL-4 and IL-13 play an important role in TH2 and B lymphocyte responses for IgE synthesis. Despite the fact that both interleukins have different actions in asthma, most of the MoAb are in development or some controlled evaluation studies are being initiated
**MoAb** **IL-13**	
**Lebrikizumab** **Anrukinzumab** **Tralokinumab**	All three MoAbs are under evaluation in controlled clinical trials. The main information found corresponds to Anrukinzumab and lebrikizumab. The former has been tested in patients with mild allergic asthma showing a slight reduction in asthmatic responses, while lebrikizumab has been administered subcutaneously to individuals with uncontrolled moderate to severe persistent asthma
**Anti-IL-9 monoclonal antibodies**	
**MEDI-528**	IL-9 (produced by TH2 lymphocytes and mast cells) has shown to have an increased expression in the airways of asthmatic individuals. Studies evaluating subjects with mild and moderate asthma suggested that MEDI-528 binds this interleukin and has an acceptable safety profile and a decrease in exacerbations and FEV1 after doing physical exercise.
**Anti-TNF-α**	
**Etanercept** **Infliximab** **Adalimumab** **Golimumab**	The four MoAb agents are still under evaluation. Etanercept is a dimeric protein that binds to free TNF-α by neutralizing it; however, it has not shown improvement in any asthmatic parameter so far. While in a single study, golimumab apparently had action on severe asthma. Infliximab (considered for moderate persistent asthma) and adalimumab (tested for severe chronic asthma) have apparently shown reduced exacerbations. Unfortunately, there are no verifying results of its efficacy and safety
**MoAb against T cells**	
**Daclizumab** **Keliximab** **Oxelumab** **KB003**	There is little information on these drugs. Inflammation of the airway is known to involve the activation of T lymphocytes, with an increase in T CD25 + cells, in concentrations of IL-2, and a chain receptor of soluble IL-2 (IL-2R). In a single study, daclizumab was confirmed to act on this receptor and appears to have activity against uncontrolled moderate-severe asthma. Apparently, Oxelumab has an effect in patients with controlled mild allergic asthma by blocking OX40. This mechanism is related to the costimulation between the dendritic cell and the T lymphocyte. On the other hand, KB003 has shown changes in FEV1 and lowered the number of exacerbations in individuals with uncontrolled moderate to severe asthma. Keliximab has only been used in the treatment of rheumatoid arthritis

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
