# Peer review of "Asthma: New Integrative Treatment Strategies for the Next Decades"

_medicina, 2020, doi:10.3390/medicina56090438_

Round 1

Reviewer 1 Report

An interesting review summarizing data related ROS and NOS role in asthma pathogenesis and its influence on its treatment.

The new GINA 2020 guidelines should be cited instead of 2019.

Few language corrections should be made.

Author Response

Dear reviewer

The authors appreciate the comments and observations of the article.

Thanks for everything

Receive a cordial greeting

We have considered the suggestions and observations.

We have corrected the data to the GINA 2020 guidelines and include the reference “Global Strategy for Asthma Management and Prevention. Global Initiative for Asthma (GINA) 2020. Available online: https://ginasthma.org/wp-content/uploads/2020/06/GINA-2020-report_20_06_04-1-wms.pdf”

Also, we have corrected some grammatical errors.

Reviewer 2 Report

This is a very comprehensive review of the literature on asthma pathophysiology, diagnostic practices, and treatment. The authors do a great job describing risk factors and known mechanisms associated with asthma development and exacerbation. They discuss the main pharmacological treatments, as well as antibody therapy and use of natural products. Overall, it is an excellent overview of the status of the science regarding this disease. 

I only have a few minor suggestions for improvement:

1) On page 2, the authors acknowledge sex differences in asthma prevalence across the lifespan. It would be appropriate to mention the known roles of sex hormones in modulating asthma phenotypes, as well as incidence in pregnancy and menopause.

2) Similarly, while genetic factors as risk factors are properly discussed, including information about known racial disparities in the disease incidence and severity will improve the manuscript.

3) When discussing natural products, it would be worthwhile mentioning the route of administration, frequency, dose, etc. if known. The authors use phrases like "the effect of the plants" (line 458) but it is unclear what they mean. 

4) line 476, the mouse strain correct name is "C57BL/6"

Author Response

Dear reviewer

The authors appreciate the comments and observations of the article.

Thanks for everything

Receive a cordial greeting

We have considered all the suggestions and observations.

1) Page 2. Line 71. We have included a new paragraph where it is mentioned about the relationship of the female gender with hormonal fluctuations during menstruation, pregnancy and menopause.

Reference 8. Jeffrey, A.Y.; Hubaida, F.;Dawn, C.N. Hormones, Sex, and Asthma. Ann. Allergy Asthma Immunol. 2018. 120, 488-494.

2) Page 6. Line 220.We have included a new paragraph mentioning racial disparity in the incidence of asthma

References 28, 29 and 30.

Forno, E.; Celedón, J.C. Health Disparities in Asthma. Am. J. Respir. Crit. Care Med. 2012, 185, 1033-1043.

Akinbami, L.J.; Moorman, J.E.; Simon, A.E.; Schoendorf, K.C. Trends in racial disparities for asthma outcomes among children 0-17 years, 2001-2010. J. Allergy Clin. Immunol. 2014, 134, 547–553.

Brewer, M.; Kimbro, R.T.; Denney, J.T.; Osiecki, K.M.; Moffett, B.; Lopez K. Does neighborhood social and environmental context impact race/ethnic disparities in childhood asthma? Health Place. 2017, 44, 86-93.

3) Page 16. Line 481. We have included a paragraph where it is mentioned that the traditional consumption of these medicinal herbs is orally and generally as aqueous extracts or tea.

Page 15. Line 472. We have also corrected the wording of the paragraph…”Mainly, the anti-asthmatic effect of the plants”

4) Page 16. Line 495. We have corrected the name of the mouse strain "C57BL/6"
